# Task-based activation and resting-state connectivity predict individual differences in semantic capacity for complex semantic knowledge

Giuseppe Rabini [1] [✉], Silvia Ubaldi[1] & Scott L. Fairhall [1]

Our ability to know and access complex factual information has far reaching effects, influencing our scholastic, professional and social lives. Here we employ functional MRI to assess the relationship between individual differences in semantic aptitude in the task-based activation and resting-state functional connectivity. Using psychometric and behavioural measures, we quantified the semantic and executive aptitude of individuals and had them perform a general-knowledge semantic-retrieval task (N = 41) and recorded resting-state data (N = 43). During the semantic-retrieval task, participants accessed general-knowledge facts drawn from four different knowledge-domains (people, places, objects and 'scholastic'). Individuals with greater executive capacity more strongly recruit anterior sections of prefrontal cortex (PFC) and the precuneus, and individuals with lower semantic capacity more strongly activate a posterior section of the dorsomedial PFC (dmPFC). The role of these regions in semantic processing was validated by analysis of independent resting-state data, where increased connectivity between a left anterior PFC and the precuneus predict higher semantic aptitude, and increased connectivity between left anterior PFC and posterior dmPFC predict lower semantic aptitude. Results suggest that coordination between core semantic regions in the precuneus and anterior prefrontal regions associated with executive processes support greater semantic aptitude.

[1] Center for Mind/Brain Sciences (CIMeC), University of Trento, Trento, Italy. [✉]email: giuseppe.rabini@unitn.it

Our brains can know that a massive black hole lies at the centre of the Milky Way or understand that trisomy of chromosome 21 is the prevalent cause of Down's Syndrome. Such knowledge necessitates the progressive amalgamation of singular concepts into increasingly complex conceptual units, culminating in a unique piece of factual knowledge. This distinctively human form of knowledge impacts multiple facets of our lives, extending beyond the scholastic and professional realm to broad aspects of the human experience, such as informing the conversations we have with others. Having a greater capacity to know and access this form of complex information may convey advantages in day-to-day life, and understanding the cortical characteristics that underlie higher or lower semantic capacity can provide insight into both variations across individuals and the neurocognitive process itself.

Semantic memory includes a broad range of knowledge that is unrelated to the experience of a specific individual. This extends from the meaning of words, to knowledge about basic concepts ('dog': a four legged domesticated mammal), to complex encyclopaedic knowledge about the world[1]. While knowledge about basic concepts is relatively uniform across individuals, quantifiable variations exist in our semantic capacity, in terms of our ability to acquire, retain and reliably access complex forms of encyclopaedic factual knowledge, as measured by standardised tests such as the general information scale of the Weschler Adult Intelligence Scale (WAIS). Here, we have used semantic aptitude as a convenience label that we operationally define as the observed performance on standardised and custom tests of general factual knowledge and does not imply that this is an unchangeable aspect of cognition that is not altered and shaped by culture or even personal proclivity.

Knowledge about singular concepts has been extensively studied in past decades. Singular concepts, often studied through object-concepts ('a hammer', 'a dog'), are thought to be represented in the brain as a conglomeration of associated semantic features with categories emerging through semantic features shared across objects[2,3]. It is known that damage to temporal lobe structures can lead to loss of conceptual knowledge, irrespective of input modality[4,5], and that these deficits can be selective for specific object categories, such as animals or tools[6–10]. Neuroimaging evidence has identified a distributed network of regions that responds more to semantically richer stimuli, extending beyond temporal lobe structures[11]. These regions largely fall within the default-mode network, encompassing the angular gyrus, dorsomedial and ventromedial prefrontal cortex, and the precuneus, as well as ventral and lateral temporal lobes. These regions are themselves sensitive to the semantic content of presented concepts[12–15] and are likely supported by representations within modality-selective cortices more tightly bound to sensorimotor experience[3,16–19].

The semantic system also encompasses regions implicated in semantic control that are thought to guide the retrieval of information relevant to the current goal or context[17,20]. During access to knowledge about singular concepts, the lateral PFC is more strongly activated when selecting between multiple competing semantic responses, or when making infrequent semantic associations, and is believed to be a key element in semantic control system[20–23]. In addition to lateral PFC, more recent models suggest that semantic control additionally involves a posterior section of the dorsomedial prefrontal cortex (dmPFC), anatomically overlapping with the supplementary motor cortex (SMA), as well as, potentially, the posterior middle temporal gyrus[17,24,25].

The way in which concepts are integrated into higher-order meaning has more recently become a topic of active research[26]. While centralised, default-mode, semantic systems may be critical in the combination of individual concepts into higher-order meaning[26,27], sentences drawn from different object domains (people, places, food, animals and objects) do selectively recruit regions beyond the default-mode system[28]. Moreover, these domain-selective representations persist when sentences draw on multiple object-domains and are associated with increased activation of the precuneus, a key default-mode hub, suggesting a possible role of centralised default-mode regions in the linking concepts across domains. At the same time, the type of semantic content retrieved about an object can interact with object-domain-selective cortices. For instance, accessing non-typical information about the geographical provenance of a famous food dish recruits regions normally associated with the processing of places[29]. Recent work comparing successful and unsuccessful access to complex factual information identified extensive semantic control regions in left lateral PFC and dmPFC involved in both successful and transiently blocked semantic access and widespread activation across default-mode and knowledge-domain-selective regions when information is successfully accessed[30]. Overall, existing work has identified a highly distributed set of cortical regions involved in access to complex semantic knowledge, incorporating conceptual representation, conceptual integration, and semantic control mechanisms.

Individual variations in semantic aptitude play out over this complex cortical landscape and may arise in control systems that guide semantic retrieval, in elements that represent knowledge, or in some combination of the two. Understanding which cortical mechanisms are associated with individual differences in semantic capacity can provide insight into the mechanisms that support access to complex factual knowledge as well as the neurocognitive strategies that confer better semantic ability. In the present fMRI study, we examine the influence of individual differences in semantic and executive capacity in the cortex. We first employ an active semantic-retrieval task, where participants access stored factual information related to 240 general-knowledge questions to identify brain regions associated with inter-individual differences in the semantic system when it is engaged in semantic access. Secondly, we confirm the role of these regions in independent data using resting-state connectivity to determine whether intrinsic differences in cortical organisation are associated with differences in semantic capacity.

## Methods

**Participants**. Seventy-three participants (20 males, mean age = 24.4 years) were pre-screened on the Information and Digit Span subscales of the Wechsler Adult Intelligence Scale 4th Edition (WAIS-IV) to facilitate selection of a sample normally distributed on these two measures (see *Session 1*, below). This approach was chosen to attain an adequate spread in the distribution of available scores and increase the sensitivity of brain/behaviour.

Forty-three participants were selected for inclusion in the fMRI study. All selected participants were right-handed native Italian speakers, with no history of neurological disorders. Two participants were excluded from the fMRI-task analysis due to within-run head movements exceeding 2.5 mm in two or more runs. For the functional resting-state analysis, no subjects were excluded, as they completed at least one run out of two with head movements lower than 2.5 mm. Thus, the final sample included a total of 43 participants (16 males, mean age = 23.4 years), 41 of which were included in the fMRI task analysis (14 males, mean age = 24.1 years). Participants gave informed consent and were reimbursed for participation (15 €/h for the MRI scanning protocol, 7 €/h for the behavioural testing). The study was approved by the Ethics Committee at the University of Trento

and was conducted in line with the declaration of Helsinki (1964, amended in 2013).

**Experimental procedure and design.** The full protocol of the present study included three testing sessions over three different days: a pre-test session delivered online, a second session combining the scanning procedure and one set of behavioural tests, and a third session to conclude the behavioural testing.

As detailed below: in *Session 1* (pre-test), participants underwent the General Information subtest and the Digit Span subtest of the WAIS (mean duration of 15 minutes); in *Session 2*, participants underwent the scanning procedure, consisting of the fMRI task (4 experimental runs of 7 min each), 2 resting-state runs (8 min each) and 2 structural acquisition sequences (6 min each). On the same day, outside the scanner, participants performed a post-scanner knowledge-verification test, the Verbal Fluency test semantic and phonetic parts[31] and the Vocabulary subscale of the WAIS. In *Session 3*, participants were administered the WAIS; then the Semantic Encoding test, with around 6 min between the encoding and retrieval phase, in which participants filled in documents and performed the Coding subscale of the WAIS. Following that, participants performed the Symbol Search and Similarities subscales of the WAIS.

Participants additionally underwent the Autobiographical Memory Interview[32] and a custom-made episodic encoding test, which are not considered in the present study.

*Stimuli - fMRI task.* Stimuli for the fMRI task consisted of 240 written questions in Italian, covering general-knowledge information drawn from four knowledge domains: People, Places, Objects and Scholastic.

Person-related questions considered facts about contemporary or historical, real or fictional, famous people (People; e.g., "What is the name of the main character in "The Lord of the Rings"?"). Place-related questions considered knowledge about geographic locations or monuments (Places; e.g., "If I am in Cappadocia, what country am I in?"). Object-related questions considered specific objects or tools (Objects; e.g., '"What is the instrument capable of measuring atmospheric pressure?"). Scholastic question related to more abstract facts and knowledge, likely acquired in an educational context (Scholastic; e.g., "In chemistry, what is the opposite of an acid?"). Questions were derived from our previous study[30]. These questions were selected to be challenging (sample reported 46% of the answers to be known) and with sufficient variation in item accuracy across participants (the mean correlation across participants between items reported known/not was $r = 0.21$). The complete list of stimuli is presented in Table S1.

Stimuli wording was constructed to create sentences of 8–12 words (mean 10.0) and were balanced across knowledge domain by number of words ($F_{(3,236)} = 1.13$, $p = 0.337$) and number of letters ($F_{(3,236)} = 1.12$, $p = 0.343$). Questions were presented one word at a time for 250 ms each, in black with the left-of-centre letter printed in red ($\approx$17% left-of-centre), in order to facilitate fixation.

In an additional *control condition*, trials consisted of the presentation of a list of 8–12 words (mean 10.0), randomly selected from a list of 90 common nouns. The order of the words in the list was randomly defined and as a result did not have any meaning at the sentence level. Presentation parameters were identical to the experimental conditions, with the distinction that each word was written in blue with a left-of-centre letter printed in green colour to facilitate fixation. 12 control trials were present in each run (48 in total).

*fMRI experimental task.* The task-based fMRI session was divided into 4 experimental runs lasting 7 min each. In each run, 15 questions for each of the four knowledge-domains and 12 trials of the control condition were presented in a pseudo-randomised event-related design (72 trials in each run). The fMRI task session thus included 288 trials in total: 240 questions (60 for each domain) and 48 control trials. Stimuli were presented using MATLAB (www.mathworks.com) and Psychtoolbox Version 3 (www.psychtoolbox.org).

Trial duration was 6 s with 2–3 s of word presentation (depending on sentence length), followed by a red fixation cross that cued the response interval. For the experimental conditions, the task of the participants was to indicate whether the answer to the question was 'known' or 'unknown' via button press. Participants were instructed to indicate 'known' only if they could access the answer in that moment and to indicate 'unknown' either if the knowledge was absent or temporary inaccessible. For the control condition, participants pressed a button at the end of stimulus presentation, when a green cross appeared. When participants responded, the cross turned black until the end of the trial.

*Post-scanner knowledge-verification test.* After the fMRI session, participants were administered a knowledge-verification test to validate the responses given in the scanner. We presented participants with 20% of the 'known' questions, as well as 20% of the 'unknown' questions. For each trial, questions were presented on the screen and participants had up to 5 s to indicate whether they knew the answer via button press. If participants indicated that they knew the answer, they were instructed to type in the response (without time limitation).

*Psychometric measures (predictors).* To investigate individual differences in cognitive abilities, we adopted different standardised and custom-made measures to be used as regressors of interest in our experimental design. Specifically, participants performed six subscales of the WAIS-IV. These were Information (general-knowledge questions designed to assess ability to acquire, retain and retrieve information), Digit Span (repeating back a series of numbers, forward and backwards, to assess verbal working memory), Vocabulary (word definitions, to assess word knowledge and verbal concept formation), Similarities (stating how two words are alike, to assess verbal reasoning and concept development), Symbol Search (locating target symbols in an array, measuring processing speed and concentration) and Coding (associating symbols with numbers and providing the appropriate symbol in a list, to assess processing speed, short term memory and concentration). Participants additionally performed a Verbal Fluency Test and a custom-made Semantic Encoding test.

*Semantic encoding test.* We developed a new behavioural test investigating encoding of semantic information. The test is composed of an encoding phase and a retrieval phase, performed about 6 min apart. In the *encoding phase*, participants listened via headphones to a list of uncommon facts and were instructed to pay attention to the facts because they would later be asked about this information.

As stimuli, we used 25 unusual facts taken from the Internet (www.fattistrani.it); e.g., *"Thomas Young has been called 'The last man who knew everything', he demonstrated the wave nature of light, he developed the theory of capillarity and, among other things, he deciphered Egyptian hieroglyphs".* We selected the facts and created a final description in order to have a main subject (e.g., *Thomas Young*) and four possible details to remember (e.g., 1) *named 'The last man who knew everything'*; (2) *demonstrated*

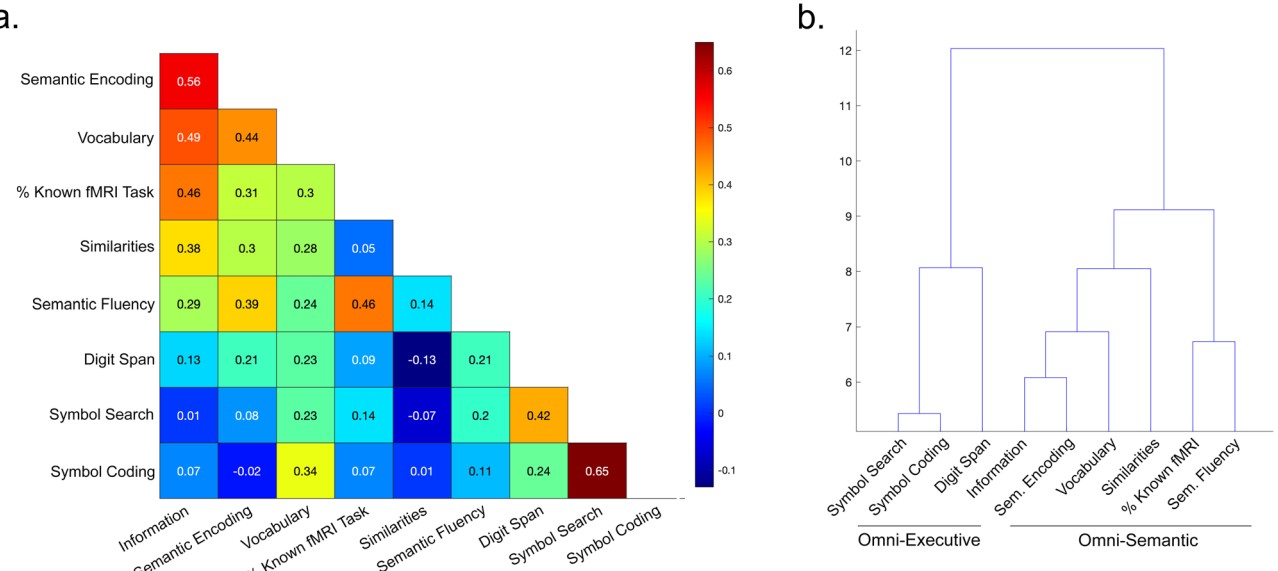

**Fig. 1 Relationship between psychometric predictors. a** Pearson's correlation coefficient of the pairwise correlation between each predictor variable.
**b** Ward hierarchical clustering analysis indicates two clusters corresponding to the semantic ('Omni-Semantic') and executive ('Omni-Executive') factors used in the primary analysis.

*the wave nature of light*; (3) *developed the theory of capillarity*; (4) *deciphered Egyptian hieroglyphs*. Participants listened to the facts recorded by a female speaker (mean duration = 16.1 s, range [14.9–17.5 s]) with a 5-s inter-trial interval. The order of presentation of the facts was randomised across participants.

In the *retrieval* phase, participants were required to verbally respond to randomly ordered, pre-defined questions for each presented fact (e.g. "What did Thomas Young, 'The Last Man Who Knew Everything', decipher?").

The full list stimuli is presented in Table S2.

**Composite measures**. To reduce multiple comparisons and focus initially on broader aspects of cognition, we collapsed like scales that cluster together across individuals (see dendrogram in Fig. 1) into two composite measures (see methods). Specifically, summary composite measures of semantic and executive capacity were constructed in the following way. Omnibus-Semantic: Information, Vocabulary, and Similarities subscales were combined and weighted according to WAIS-IV criteria (which is non-linear; Wechsler[33]). A weighted average was then performed including this measure, the percentage of answers known from the fMRI test, the Semantic Encoding test and the Verbal Fluency tests (specifically, the composite Information/Vocabulary/Similarity measure was weighted by 3 to reflect the number of initial tests, while the other scales were weighted by 1). Omnibus-Executive: Symbol Search and Coding subscales were summed and weighted according to WAIS-IV criteria. This measure was then z-transformed, multiplied by 2, summed with the Digit Span score and divided by 3.

**MRI scanning parameters**. Data were acquired at the Center for Mind/Brain Sciences (CIMeC) of the University of Trento, with a Prisma 3 T scanner (Siemens AG, Erlangen, Germany) and using a 64-channel head coil.

In the scanner, we presented the visual stimuli (written words) through a mirror system connected to a 42" LCD monitor (MR-compatible, Nordic NeuroLab) positioned at the back of the magnet bore.

Functional images were acquired using echo planar (EPI) T2*-weighted scans. Acquisition parameters were: repetition time

(TR) of 2 s, an echo time (TE) of 28 ms, a flip angle of 75°, a field of view (FoV) of 100 mm, and a matrix size of 100 × 100. Total functional acquisition consisted of 888 volumes, for the four experimental runs, and 470 volumes for the two 8-min-long resting state runs, each of 78 axial slices (which covered the whole brain) with a thickness of 2 mm and gap of 2 mm, anterior-/posterior commissure aligned. Two 6-min-long high-resolution (1 × 1 × 1 mm) T1-weighted MPRAGE sequences were also collected (sagittal slice orientation, centric phase encoding, image matrix = 288 × 288, field of view = 288 mm, 208 slices with 1-mm thickness, repetition time = 2290, echo time = 2.74, TI = 950 ms, 12° flip angle).

**fMRI data analysis**

*Semantic-access task.* Data were analysed and preprocessed with SPM12 (http://www.fil.ion.ucl.ac.uk/spm/). The first four volumes of each run were dummy scans. All images were slice-time corrected, realigned to correct for head movement, normalised to MNI space and smoothed using a 6 mm FWHM isotropic kernel. Before computing the General Linear Model, the four runs were concatenated to avoid empty parameters in one or more conditions. Nine trial types were modelled for the recall event: the two different response types (Known, Unknown) for each of the four knowledge domains, plus the control condition. Subject-specific parameter estimates (β weights) for each of these nine types were derived from the GLM. The six head-motion parameters were included as additional regressors of no interest. Group-level analysis was performed in one random-effects GLM and two separate whole-brain regressions, with the predictor variables Omni-semantic and Omni-executive as regressors.

*Resting-state functional connectivity.* Data were analysed using the CONN toolbox for MATLAB v17a[34], available at https://www.nitrc.org/projects/conn. Six participants had one session excluded due to excessive head movement (<2.5 mm). The same pre-processing pipeline was used as in the task-based fMRI analysis, with the addition of functional outlier detection (Artifact Detection Toolbox-based identification of scans for scrubbing; https://www.nitrc.org/projects/artifact_detect/), bandpass filtering (0.01–0.1 Hz) and inclusion of white matter and CSF time series

as regressors of no interest. Functional connective indices (Fisher-transformed correlation coefficients) were then calculated between (a) regions of interest (ROI) and all voxels and (b) ROI to ROI (see results). At the group level, intrinsic connection strengths were assessed via second-level random-effects analysis and individual differences were assessed by regressing connectivity indices with the predictors of interest across participants.

*Region of interest definition.* ROIs were defined at the group level using the MATLAB toolbox MarsBar[35]. (1) Task-based fMRI analyses were used to form seed regions for resting-state analysis. Specifically, four ROIs were created based on regions showing individual differences as a function of the predictors of interest (omni-semantic and omni-executive; see Results section). ROIs were defined as the union between 5 mm radius spheres centred at the peak of the effect and voxels exhibiting significant ($p < 0.01$) modulation for the predictor of interest. (2) We used the significant clusters arising from the univariate analysis of the fMRI task on knowledge-domain selectivity to create category-selective ROIs for each of the four knowledge domains. Specifically, for each knowledge domain, knowledge-domain-selective ROIs were defined as the union of a 5-mm-radius sphere around the six most significant category-selective peaks (i.e. the local maxima as reported in Table 2), and voxels exhibiting significant ($p < 0.001$) selectivity for the domain at the group level.

**Statistics and reproducibility**. We implemented random-effects analyses across individual participants in both the task-based ($N = 41$) and resting-state ($N = 43$) analyses with the statistical reliability of the reported effects calculated using parametric estimates of the magnitude of the difference between conditions proportional to the amount of variability in this effect seems across participants (*t*-tests, *F*-tests). Within sample, the role of cortical regions in predicting individual variations in semantic access during task were largely replicated in the functional connectivity analysis of the separate resting-state dataset from those same individuals.

**Reporting summary**. Further information on research design is available in the Nature Portfolio Reporting Summary linked to this article.

## Results
### Behavioural results
*fMRI task.* Participants reported that they knew the facts on 46.67% of the trials, on average. Post-scanner testing confirmed the veracity of responses given in the scanner, with participants able to provide the correct answer for 93.5% of known facts and only 12.4% of unknown facts.

Response type differed across knowledge domains, as revealed by a repeated-measures ANOVA ($F_{(2.4,95.1)} = 6.58$, $p < 0.001$, Greenhouse-Geisser corrected). Specifically, post-hoc tests indicated that the mean number of known facts for People (25.6/60, 42.7%) and places (26.0/60, 43.4%; no significant differences)

were lower than known facts for Object (30.1/60, 50.1%; vs. people, $t_{(40)} = 3.18$, $p = 0.011$, vs. places, $t_{(40)} = 2.87$, $p = 0.029$) and Scholastic domains (30.3/60, 50.5%; vs. people, $t_{(40)} = 3.39$, $p = 0.006$; vs. places, $t_{(40)} = 3.08$, $p = 0.015$).

Repeated-measures ANOVA revealed reaction times (RTs) differed as a function of response type ($F_{(1,40)} = 28.88$, $p < 0.001$), with faster RTs for known (1285 ms) than unknown facts (1428 ms). RT differences were also evident as a function of knowledge domain ($F_{(2.4, 97.5)} = 55.66$, $p < 0.001$, Greenhouse-Geisser corrected). Responses for People (1191 ms) were faster than Places (1324 ms, $t_{(40)} = 5.53$, $p < 0.001$), Objects (1448 ms, $t_{(40)} = 10.71$, $p < 0.001$) and Scholastic (1463 ms, $t_{(40)} = 11.31$, $p < 0.001$). RTs for Place trials were faster than Objects ($t_{(40)} = 5.18$, $p < 0.001$) and Scholastic ($t_{(40)} = 5.78$, $p < 0.001$) trials, which did not differ significantly from one another. The interaction between response type and knowledge domain was also significant ($F_{(3,120)} = 13.60$, $p < 0.01$), reflecting smaller average differences between known and unknown responses for people (<1 ms), compared to other categories (places: 120 ms; objects: 203 ms; scholastic: 251 ms; *p*-values < 0.01), as well as smaller known-unknown reaction-time differences for place than for scholastic knowledge-domains ($p = 0.005$).

*Psychometric predictors.* Participants performed different cognitive tests involving standardised measures (from the WAIS) and custom-made measures (see methods). The following data relates to the whole sample ($n = 43$), given that resting-state data covers all recruited subjects.

Overall, our sample showed higher average scores than the WAIS-IV normalised mean (10) in most of the subscales, with the Information and Digit Spans most closely approximating the population norms. The complete descriptive statistics of the single experimental covariates are shown in Table 1, and the relationship between individuals' performances across tests is shown in Fig. 1.

The questions used in the fMRI study are not designed to be a standardised test and have not been validated across a wide range of samples (gender, age). However, within our northern Italian population, the scale demonstrated reasonable convergent validity with Verbal subtests of the WAIS (Information, Vocabulary, Semantic Fluency, Similarities; cf. Fig. 1), that was maximal with the Information scale ($r = 0.46$). At the same time, performance on the fMRI task demonstrated discriminate validity with respect to the Performance tests (Digit Span, Symbol Search, Symbol Coding). Collectively, this suggests that the fMRI task has an acceptable construct validity for this population.

**Successful access to semantic knowledge**. To identify the neural response related to successful accessing of stored semantic knowledge, we compared trials of known versus unknown facts (Fig. 2). Results reveal an extensive bilateral network of cortical regions previously implicated in semantic access: the precuneus, vmPFC and dmPFC, the cingulate cortex, the bilateral ventromedial temporal lobe encompassing the hippocampi, the parahippocampal cortex, the retrosplenial cortex and the fusiform

**Table 1 Descriptive statistics of individual psychometric measures.**

| | Information | Digit Span | Vocabulary | Similarities | Symbol Search | Coding | Semantic Fluency | Semantic Encoding | % Known fMRI Task | Omnibus SEMANTIC | Omnibus EXECUTIVE |
|---|---|---|---|---|---|---|---|---|---|---|---|
| Mean | 11.16 | 9.98 | 13.35 | 13.30 | 12.88 | 13.37 | 48.96 | 56.93 | 46.20 | 0.00 | 0.00 |
| std | 3.02 | 2.45 | 2.49 | 2.60 | 1.89 | 2.65 | 10.11 | 13.23 | 12.71 | 0.78 | 0.85 |
| min | 5.00 | 4.00 | 9.00 | 3.00 | 9.00 | 8.00 | 31.50 | 16.00 | 17.50 | −1.65 | −2.07 |
| max | 17.00 | 15.00 | 18.00 | 18.00 | 18.00 | 19.00 | 69.50 | 76.00 | 74.17 | 1.95 | 1.80 |

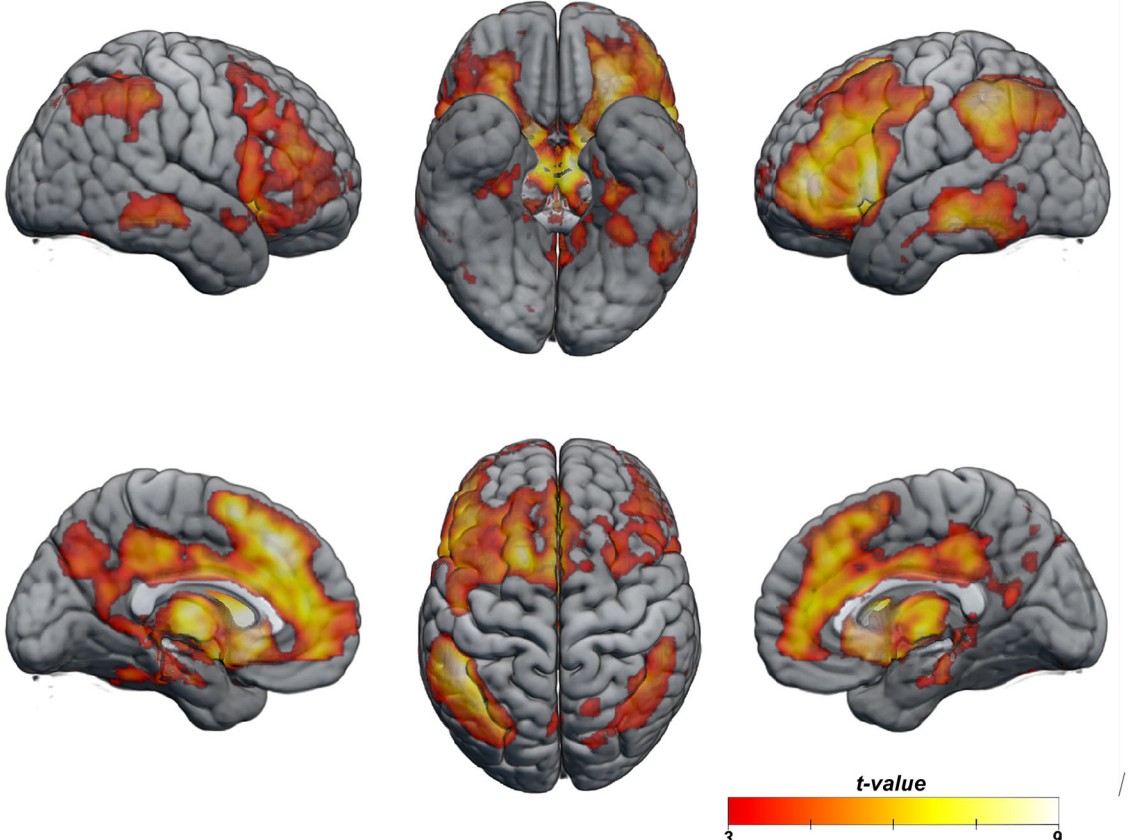

**Fig. 2 Successful access to complex factual knowledge.** Statistical parametric map of cortical regions showing a significantly greater response for known trials versus unknown trials (initial voxel-wise threshold, $p < 0.01$, cluster-corrected at $p < 0.05$).

gyrus, bilateral angular gyrus, bilateral posterior portions of the middle temporal gyrus and widespread recruitment of lateral PFC, extending from premotor cortex to anterior sections of the PFC, with peak activation in the inferior frontal gyrus (IFG). While effects were present in both hemispheres, they were more pronounced in the left hemisphere.

In addition to regions involved in semantic processing, the comparison between known and unknown facts highlighted the prominent recruitment of regions classically involved in reward circuitry: the orbitofrontal cortex, caudate nucleus and nucleus accumbens (see also Rabini et al.[28]).

**Differential distribution of knowledge-domains across the brain.** The stimuli used in the present experiment reference different domains of factual knowledge (People, Places, Objects and Scholastic), allowing the identification of regions sensitive to the semantic content of the factual information. To identify knowledge-domain-selective voxels, we considered both known trials and unknown trials and contrasted each single category to the average of the remaining three (e.g. [People > Places, Objects, Scholastic]). Known and unknown trials were used for consistency with previous work (Ubaldi et al., 2022[30]) and results did not qualitatively differ if only known trials were used.

Results indicate pronounced selectivity for knowledge about people in both hemispheres: bilateral angular gyrus, bilateral lateral and anterior portion of the ATL, bilateral fusiform gyrus and bilateral amygdala (Fig. 3, Table 2). To assess whether these knowledge-domain-selective regions reflect intrinsically coupled networks, we asked whether those domain-sensitive regions are functionally coupled at rest. Resting-state functional connectivity within task-based category-selective regions of interest (ROIs; see

methods) showed that most of the regions belonging to a specific knowledge domain are functionally clustered together during rest. Figure 3b shows the hierarchical clustering of the connectivity profile between knowledge-domain-selective regions during rest using parametric multivariate statistics with familywise error controlled via false-discovery rate[36]. Generally, domains can be seen to cluster as a function of knowledge domain. Two exceptions were evident: a region of the retinotopic visual cortex selective for the scholastic knowledge-domain clustered with a region of the lateral fusiform gyrus and the object-selective parahippocampal region clustered with place-selective regions that included adjacent PPA.

**Individual differences in activation during semantic access.** To investigate individual differences in regional activation during the semantic knowledge-access task, we asked whether the brain activity specifically related to known facts (fMRI contrast [Known > Unknown]) could be predicted by the two omnibus measures defining semantic and executive aptitude (Fig. 4). Specifically, at each voxel of the brain, we used difference in individuals' scores on the omnibus semantic or executive scale to predict the magnitude of the difference between the BOLD-derived beta response for known versus unknown trials. Here, a more lenient, initial, voxel-wise threshold of $p < 0.01$ was employed to maximise sensitivity at the voxel level while identifying relatively large clusters at the corrected inferential level. This lower threshold allows us to detect voxels whose involvement is more subtle while keeping statistics at the inferential level consistent at $p < 0.05$ corrected. However, this lower initial threshold may be associated with an increased probability of false positives[37,38] and it is therefore important that results within the identified regions are validated and verified in the subsequent

**Table 2 Domain-sensitive brain regions.**

| Category | Region | Cluster | | | | |
|---|---|---|---|---|---|---|
| | | $p_{(FWE-cor)}$ | Extent | $p_{(FWE-cor)}$ | T | x,y,z(MNI) |
| People | Precuneus | <0.001 | 3515 | <0.001 | 14.98 | 4, −52, 22 |
| | vmPFC | <0.001 | 20338 | <0.001 | 13.96 | 4, 62, −10 |
| | Right latATL | | | <0.001 | 12.96 | 62, −8, −20 |
| | Right Temp. Pole | | | <0.001 | 12.20 | 44, 10, −34 |
| | Right Fusiform | | | <0.001 | 10.49 | 38, −48, −22 |
| | dmPFC | | | <0.001 | 9.86 | 4, 60, 22 |
| | Right Amygdala | | | <0.001 | 9.80 | 20, −4, 16 |
| | Right latPFC | | | <0.001 | 8.70 | 46, 14, 32 |
| | Right dlPFC | | | <0.001 | 8.56 | 28, 28, 46 |
| | Right AG | | | <0.001 | 8.51 | 48, −60, 28 |
| | Left latATL | <0.001 | 4564 | <0.001 | 12.85 | −60, −8, −16 |
| | Left Amygdala | | | <0.001 | 10.39 | −18, −6, −16 |
| | Left ATL | | | <0.001 | | −44, 6, −36 |
| | Left Fusiform | <0.001 | 3590 | <0.001 | 10.44 | −36, −44, −22 |
| | Left AG | | | <0.001 | 7.26 | −44, −64, 36 |
| Places | Right RSC | <0.001 | 9097 | <0.001 | 28.73 | 16, −56, 18 |
| | Left RSC | | | <0.001 | 25.89 | −14, −60, 18 |
| | Left PPA | | | <0.001 | 18.73 | −28, −40, −12 |
| | Right PPA | | | <0.001 | 17.92 | 28, −36, −12 |
| | Left TOS | | | <0.001 | 16.98 | −36, −82, 30 |
| | Left superior parietal | | | <0.001 | 9.74 | −4, −68, 50 |
| | Right Hippocampus | | | <0.001 | 9.49 | 22, −18, −22 |
| | Left Hippocampus | | | <0.001 | 7.33 | −20, −24, −22 |
| | Right TOS | <0.001 | 1369 | <0.001 | 13.51 | 38, −78, 36 |
| | Right Premotor | <0.001 | 853 | <0.001 | 9.78 | 26, 16, 46 |
| | Left Premotor | <0.001 | 674 | <0.001 | 7.36 | −24, 10, 52 |
| | Right pCing | <0.001 | 295 | 0.001 | 5.70 | 6, −36, 40 |
| | Right pITG | 0.017 | 153 | 0.034 | 4.88 | 58, −42, −12 |
| Objects | Left IFS | <0.001 | 345 | <0.001 | 9.53 | −42, 34, 12 |
| | Left pMTG | <0.001 | 1006 | <0.001 | 9.41 | −48, −60, −4 |
| | Left pITG | | | <0.001 | 8.85 | −48, −46, −18 |
| | Left Parahipp.l | | | <0.001 | 854 | −32, −28, −22 |
| | Left SupMargG | <0.001 | 359 | 0.010 | 8.49 | −60, −28, 38 |
| | Left OFC | 0.019 | 149 | .0008 | 5.29 | −28, 38, −12 |
| Scholastic | Left IFG parsTriang | <0.001 | 2972 | <0.001 | 9.63 | −52, 34, 0 |
| | Left IFG parsOperc | | | <0.001 | 7.61 | −52, 14, 14 |
| | Left pSTS | <0.001 | 1991 | <0.001 | 9.53 | −52, −38, 0 |
| | Left Visual | <0.001 | 4503 | 0.001 | 5.83 | −12, −88, −4 |
| | Left SMA | <0.001 | 1274 | 0.001 | 5.77 | −4, 12, 58 |
| | Left Premotor | <0.001 | 382 | 0.002 | 5.61 | −40, 6, 46 |
| | Right dlPFC | 0.020 | 147 | 0.130 | 4.63 | 34, 38, 32 |
| | Right pSTS | <0.001 | 302 | 0.285 | 4.40 | 46, −34, −8 |

Significant clusters are reported separately for each selected semantic category. For large clusters, local maxima (>20 mm apart) are listed separately.

resting state analysis (see next section). Results indicate decreased recruitment of posterior dmPFC during semantic access in individuals with higher semantic aptitude ($MNI_{xyz} = −2, 32, 54$; extent = 510; $p_{FWE-corrected} = 0.01$ cluster-level). By contrast, individuals with higher omnibus-executive indexes showed larger relative responses for known trials in a prefrontal cluster (extent = 1608; $p_{FWE-corrected} < 0.00001$ cluster-level), peaking in the left anterior PFC (peak $t = 4.70$, $MNI_{xyz} = −24, 60, 18$) and extending into vmPFC (local maximum: $t = 4.26$; $MNI_{xyz} = 4, 66, 6$) and the right anterior PFC (local maximum $t = 3.59$, $MNI_{xyz} = 22, 60, 18$), as well as a second cluster in the precuneus (extent = 777; $p_{FWE-corrected} < 0.001$ cluster-level; $MNI_{xyz} = 0, −60, 28$; peak $t = 3.73$).

Follow-up analysis of the contribution of the individual subcomponents of the omni-executive measure showed that digit span strongly predicted activation in both the anterior prefrontal cluster ($MNI_{xyz} = −24, 60, 18$; extent = 3063; $p_{FWE-corrected} < 0.000001$ cluster-level) and the precuneus ($MNI_{xyz} = 0, −60, 28$; extent = 669; $p_{FWE-corrected} = 0.002$ cluster-level). Correlation with Coding or

Symbol Search subcomponents of the omni-exec measure did not approach significance (all corrected $p$-values > 0.5).

To address whether activation within regions showed sensitivity to semantic content, a pre-planned ROI analysis of individual differences in omni-executive and omni-semantic measures was performed. This revealed no modulation within any of the four domain-selective networks ($p$-values > 0.05, uncorrected).

**Resting-state functional connectivity from left lateral PFC is predicted by semantic aptitude**. The preceding analysis indicated that, during a semantic processing task, individual variations in semantic or executive aptitudes can predict activity in dorsal, medial and anterior PFC and the precuneus. To confirm the role of these regions in variations in capacity for complex factual knowledge, we assessed whether individual differences are additionally reflected in intrinsic cortical organisation. Specifically, in an independent set of data we determined whether the resting-

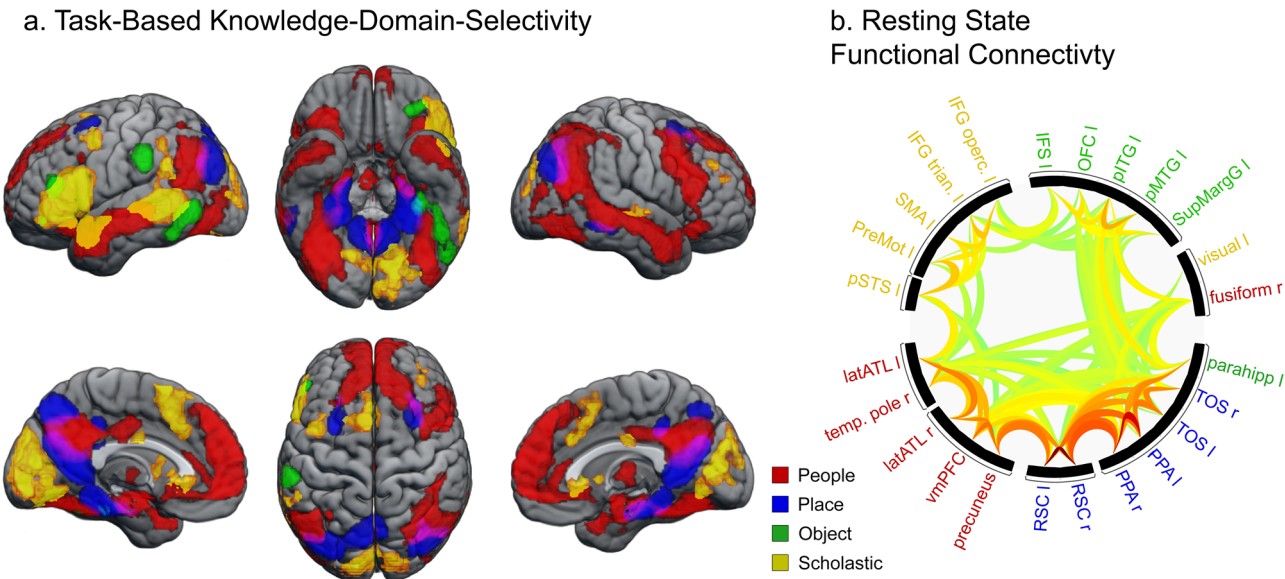

a. Task-Based Knowledge-Domain-Selectivity

b. Resting State Functional Connectivty

People
Place
Object
Scholastic

**Fig. 3 Domain-selective networks. a** Statistical parametric map of cortical regions showing knowledge-domain selectivity. For each of the four knowledge domains, the response for that domain was contrasted with the average of the other three (initial voxel-wise threshold, $p < 0.001$, cluster-corrected at $p < 0.05$). **b** Radial plot of resting-state positive functional connectivity between the six most selective nodes of each domain-selective network. Colour-code indicates the t-values of individual connection strength and solid black borders indicate significant clusters of connected regions ($p < 0.05$, FDR corrected).

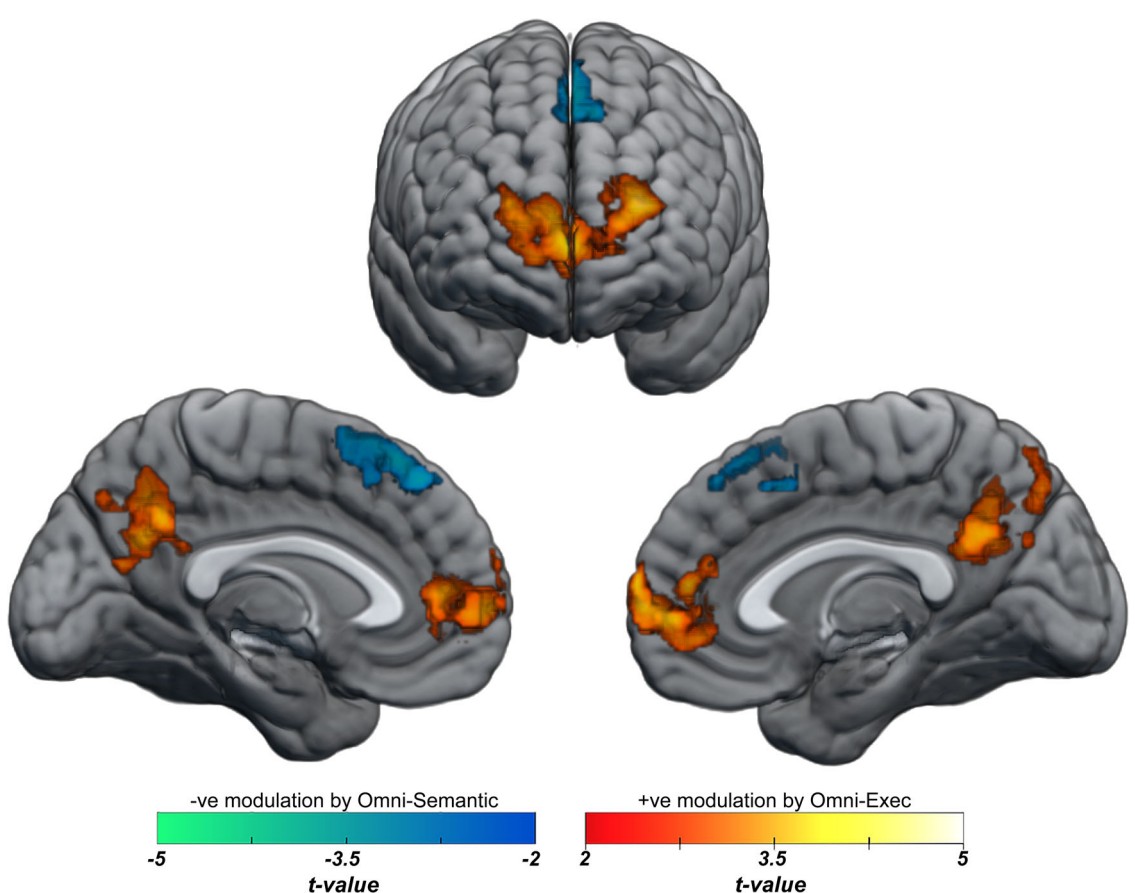

-ve modulation by Omni-Semantic

+ve modulation by Omni-Exec

-5        -3.5        -2        2        3.5        5
*t*-value                      *t*-value

**Fig. 4 Task-based individual differences in access to complex factual knowledge.** Omni-executive scores positively predict task-based (known>unknown) activation in anterior PFC, vmPFC and the precuneus. Omni-semantic scores negatively predict task-based fMRI activation in dmPFC. (Initial voxel-wise threshold, $p < 0.01$, cluster-corrected at $p < 0.05$. All clusters were >500 voxels in extent).

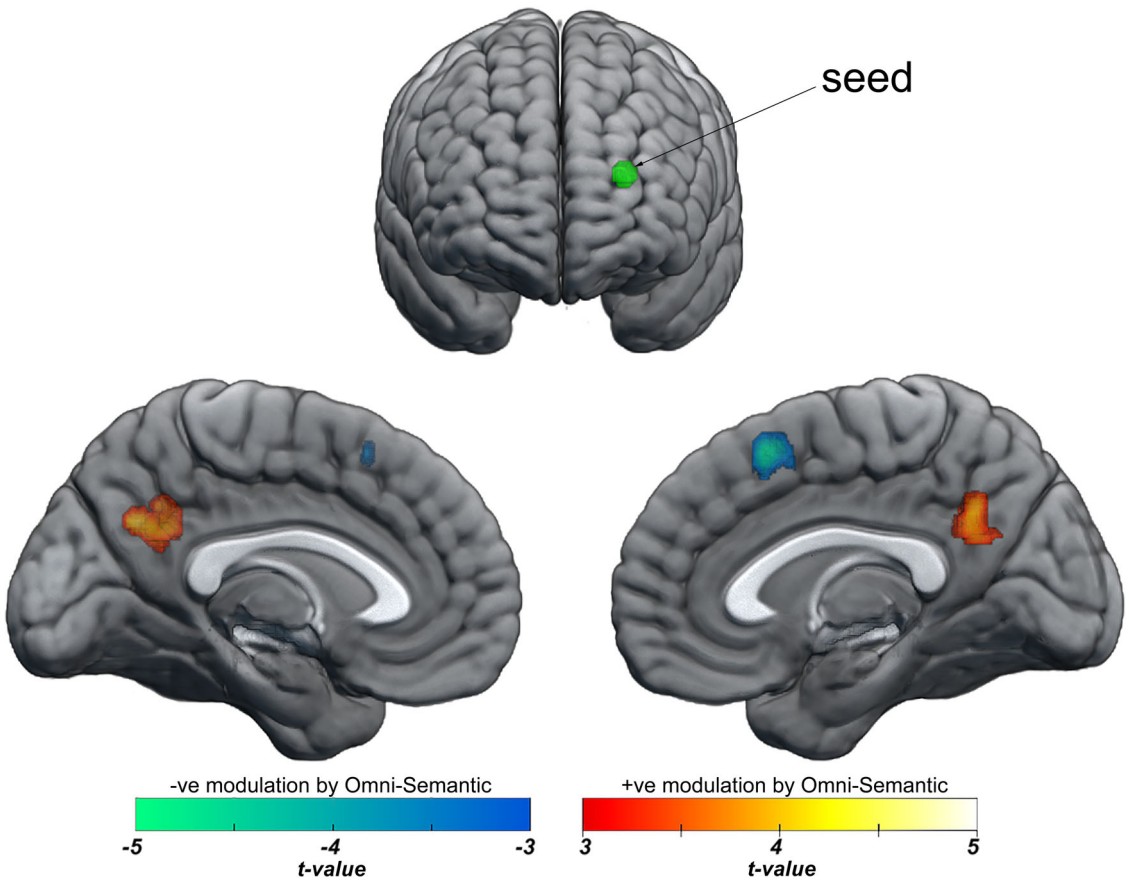

**Fig. 5 Effect of Individual differences in the omni-semantic measure on resting-state functional connectivity from the left anterior PFC seed (green).** The omni-semantic measure predicts increased connectivity between left anterior PFC and the precuneus and decreased connectivity between left anterior PFC and the posterior dmPFC.

state functional connectivity profile of the identified regions also varied as a function of semantic aptitude.

Seed-based analysis indicated that higher semantic aptitude is associated with increased connectivity between the left anterior PFC and the precuneus ($MNI_{xyz} = -10, -60, 30$; Extent: 318; $p < 0.00001$ cluster-corrected; Fig. 5). By contrast, higher semantic aptitude was associated with decreased resting-state connectivity between the left anterior PFC and the posterior dmPFC ($MNI_{xyz} = 4, 16, 56$; Extent: 120; $p = 0.007$ cluster-corrected). Both findings survived additional corrections for multiple comparisons across the six seed-ROIs analysed.

No significant modulation of connectivity strengths from seeds in vmPFC, right anterior PFC, vmPFC, posterior dmPFC or the precuneus was present.

To assess the individual contributions of the composite omni-semantic predictor, we now consider the influence of constituent tests on connectivity from the left anterior PFC separately. Connectivity to a cluster in the precuneus was also predicted by the individual Information subscale ($MNI_{xyz} = 12 \ -52 \ 30$; extent = 404; $p < 0.000001$ cluster-corrected) and the Similarities subscales ($MNI_{xyz} = -8 \ -50 \ 22$; extent = 516; $p < 0.000001$ cluster-corrected). The Similarities subscale additionally predicted positive connectivity with the right angular gyrus ($MNI_{xyz} = -42 \ -64 \ 24$; extent = 331; $p < 0.000001$ cluster-corrected), vmPFC ($MNI_{xyz} = -4 \ 64 \ 0$; extent = 516; $p = 0.00006$ cluster-corrected), right lateral mid-MTG ($MNI_{xyz} = 58 \ -6 \ -20$; extent = 115; $p = 0.023$ cluster-corrected) and negatively correlated with the left opercular IFG ($MNI_{xyz} = -46 \ 4 \ 12$; extent = 161; $p = 0.001$ cluster-corrected), right middle frontal gyrus ($MNI_{xyz} = 36 \ 34 \ 28$; extent = 110; $p = 0.012$ cluster-corrected) and the dmPFC ($MNI_{xyz} = 10 \ 8 \ 42$;

extent = 282, $p = 0.00001$). No other individual measures — Semantic Encoding, Vocabulary or Semantic Fluency — predicted connectivity from the left anterior PFC (all $p$-values > 0.5 corrected) and a negative modulation by performance in the scanner task of connectivity between the left anterior PFC and the dmPFC, did not survive correction for multiple comparisons ($p = 0.053$, corrected).

Omnibus measure of executive function, by contrast, predicted negative connectivity of left anterior PFC to two right pre/post-central clusters ($MNI_{xyz} = 36 \ -16 \ 68$, extent = 162, $p = 0.001$ cluster-corrected, $MNI_{xyz} = 36 \ -26 \ 52$, extent = 143, $p = 0.003$ cluster-corrected) and right parietal lobe ($MNI_{xyz} = 24 \ -66 \ 50$, extent = 83, $p = 0.047$ cluster-corrected).

No other comparisons approached significance.

**Left anterior PFC is linked to multiple functional networks.** The preceding analysis showed that during rest, the functional connectivity profile of the left anterior PFC is related to individual variations in semantic processing aptitudes. To more fully characterise the connectivity pattern of this cortical region, we quantified its connectivity profile to previously described resting state functional networks. Figure 6 shows that the left anterior PFC is connected both to key nodes of the default-mode network (precuneus, vmPFC, angular gyri and lateral middle temporal lobes) and to dorsolateral prefrontal regions associated with the multiple-demand network.

## Discussion

In this work, we sought to identify the neural substrates that distinguish among individuals with better or worse capacities for

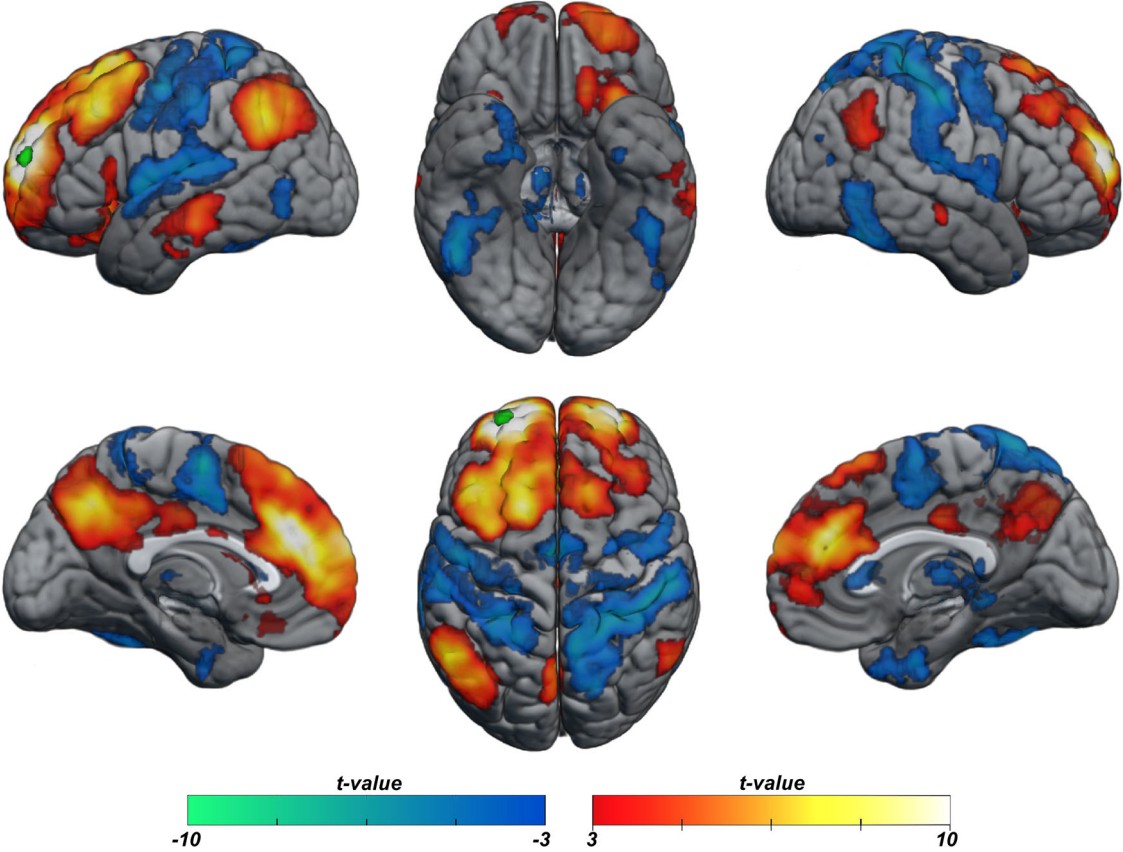

**Fig. 6 Intrinsic functional connectivity from left anterior PFC seed (green) to the whole brain.** (Initial voxel-wise threshold, $p < 0.001$, cluster-corrected at $p < 0.05$).

semantic knowledge. We observed that, during successful access to complex factual information, individuals with higher executive capacity relied more strongly on the anterior PFC, as well as on vmPFC and the precuneus. This functional pattern was validated by convergent evidence from independent resting-state data, which showed that connectivity between the left anterior PFC and the precuneus reliably predicted semantic aptitude. By contrast, increased activation of posterior dmPFC was associated with lower semantic aptitude and resting-state analysis indicated that reduced connectivity between this region and left anterior PFC predicted higher semantic aptitude.

As a preliminary step, we identified brain regions that responded more strongly during trials in which the participant reported that the fact was known (and immediately accessible), compared to trials in which the knowledge could not be accessed. This contrast was chosen to better control for broad effects of task and focus on aspects of cognition most closely associated with semantic access. Analysis of reaction times suggests that general effects of cognitive load were absent from this contrast, as RTs were longer in unsuccessful access trials. While these RT differences make it less likely that other non-specific factors, such as general memory search, are contributing to this contrast, such contributions cannot be excluded. Results revealed an extensive set of brain regions involved in successful semantic access. These included regions implicated in semantic representation: precuneus, vmPFC and dmPFC, ventromedial temporal cortices, the angular gyri and middle temporal gyri. The selective involvement of these regions in access to stored knowledge is consistent with the retrieval of representations of relevant information (see also ref. [30]). Additionally, successful access was associated with increased recruitment of regions implicated in semantic control,

the lateral prefrontal cortex, extending from premotor cortex to anterior sections of the PFC and peaking in the left inferior frontal gyrus, as well as the posterior dmPFC. The pattern of activation across lateral PFC was notably more distributed than seen in studies with single concepts, consistent with increased executive and semantic control demands associated with the processing of, and access to, more complex factual information. In addition to control and representation regions, we also saw that successful access was associated with the recruitment of regions frequently associated with reward processes: the orbito-frontal cortex and subcortical regions, including the caudate nucleus and nucleus accumbens. The reason for the involvement of reward circuitry in successful semantic retrieval is uncertain but may reflect the feeling of accomplishment associated with knowing the answer to a challenging question. Importantly, we did not see evidence that the extent of recruitment in these regions correlated with overall semantic aptitude.

We additionally identified regions that were sensitive to the knowledge domain of a fact. Consistent with previous work[28,30], we observed pronounced sensitivity to different domains of knowledge. Extending from previous research, here we quantified the resting-state connectivity between key nodes of the four knowledge-domain-selective networks to assess whether they formed functional networks. In the absence of any task, these knowledge-domain-selective regions were seen to intrinsically cluster into networks reflecting their knowledge-domain selectivity. There were two exceptions. A person-selective region anatomically consistent with the right FFA clustered with a region of the visual cortex more active for the scholastic knowledge domain. This latter region was centred on the calcarine sulcus and, while some studies suggest these regions encode conceptual

information such as object size[39], it is possible that this cluster reflects unanticipated differences in the visual processing of stimuli associated with the scholastic domain. The second exception was connectivity between object-selective parahippocampal gyrus, which clustered and place-selective network, which included adjacent PPA. These results confirm the importance of knowledge domain in the cortical representation of semantic knowledge and indicate that representational nodes interact to form network-level representations.

When accessing stored factual knowledge, those individuals with higher omnibus-executive scores and, in particular, verbal working memory capacity, rely more strongly on the anterior PFC. This anterior PFC cluster peaked in the left superior frontal sulcus and extended into the middle frontal gyrus and the frontal pole, encompassing regions associated with the retrieval of information from episodic memory[40], as well as regions of the middle frontal gyrus associated with verbal working memory monitoring[41]. It has been proposed that the anterior PFC plays a particular role in memory and attention operations related to managing internal thought processes[42,43] and that this region is recruited more strongly during tip-of-the-tongue states, when the mind frenetically seeks to access a known piece of factual information that is temporarily blocked[30,44]. Resting-state functional connectivity analysis provided further insight into the potential role of this region. On the one hand, the left-lateralised peak of the anterior PFC cluster is closely coupled to left hemispheric dorsolateral prefrontal regions implicated in a broad range of executive processes and forming a key element of the multiple-demand network[45,46]. On the other hand, this region was simultaneously connected with default-mode regions involved in internalised cognition, including semantic processes[11,47]. This common connectivity between executive and internalised cortical networks makes left anterior a plausible candidate interface between these two systems. Collectively, these results are consistent with a role of the anterior PFC in memory-related executive processes that may aid access to complex semantic information.

During the knowledge-access fMRI task, greater executive capacity was additionally associated with increased activation of the precuneus. The precuneus plays a pivotal role in semantic processing and is one of the most reliably recruited regions in semantic tasks[11]. Within this region, neural and semantic similarity spaces converge for object concepts[14], selective univariate responses are present for person- and animal-related concepts[48,49], and responses increase when object concepts must be combined across semantic domains within a sentence[28]. At the same time, this region is strongly associated with default-mode processing and inversely related to working memory or executive processes. For these reasons, it may be parsimonious to assume that variations of the activation of this region associated with inter-individual differences in executive capacity are mediated by processes occurring within the anterior PFC.

The resting-state data supports this possibility. In the absence of any task, connectivity between the left anterior PFC and the precuneus is predicted by individual differences in semantic ability. This finding not only provides convergent evidence for the role of anterior PFC in individual differences in semantic capacity but also provides further insight into the mechanisms. While those with higher working memory capacity recruit anterior PFC more strongly during a semantic task, the intrinsic link between this brain region and core elements of the classic semantic system also predicts semantic aptitude. Collectively, these results suggest that adaptive semantic processing involves the interplay between control components in the anterior PFC that facilitate the guided retrieval and integration of the progressive pieces of relevant

information and representational hubs in the precuneus, which may aid in the linking together of different representations spread across the cortex[28].

A different pattern was evident within the posterior dmPFC, a region reliably associated with semantic control processes[24]. Stronger activation of dmPFC during access to complex factual knowledge was associated with reduced semantic capacity, suggesting that reliance on forms of semantic control computed in this region reflect a less-adaptive strategy in accessing complex knowledge. However, it is unclear whether this result reflects specifics of the task, potentially compensatory strategies resulting from a lower semantic capacity, rather than an underlying cause of lower semantic capacity. Functional connectivity analysis of resting-state data provides some resolution to this ambiguity. In the absence of any task, higher semantic performance is associated with a relative decoupling of functional connectivity between the posterior dmPFC and the left anterior prefrontal cortex. This suggests that dissociations between these two cortical regions and the relative forms of semantic control they perform is an underlying contributor to enhanced semantic aptitude.

Previous work has addressed functional connectivity and individual differences as it relates to the processing of individual concepts. While task-based fMRI was not measured, Wei and colleagues[50] found that variations in the magnitude of the resting state oscillation in posterior MTG correlated with picture naming, association and feature-to-object matching performance for single-object concepts, a finding that did not extend to the retrieval of complex factual information in the present study. Vatansever and colleagues[51] did not report direct correlations between task-based activation and individual differences in semantic aptitude but did observe that decoupling between control and default-mode systems was predictive of performance in more challenging semantic tasks (e.g. identifying weak rather than strong semantic associations between items) and that this was associated with increased activation of posterior dmPFC when reading meaningful versus non-meaningful sentences. These results are in contrast to those of the present study, which highlight the importance of left anterior PFC and the precuneus in greater semantic aptitude. These differences may arise in part due to the scanning task used in the present study, which required access to relatively complex general-knowledge information, as well as measures used to quantify semantic aptitude. Specifically, studies emphasised measures that test the naming or associations between singular concepts, while the present study predominantly used tests of general knowledge and higher-level meaning.

These divergent findings highlight potential considerations relating to the measures used to quantify 'semantic aptitude'. In the present study we use a combination of standard psychometric measures and task performance as a proxy for semantic aptitude, but it should be noted that these indices do not reflect pure measurements of the underlying cognitive process and are known to be biased as a result of cultural background and socioeconomic status[52,53]. Interpretation of the present results should be made within the limitations of the psychometric scales used to quantify semantic aptitude. At the same time, it should be noted that our sample reflects a culturally and ethnically homogenous population and the external validity of the present results to differing populations remains untested and, even within this population, additional relationships between aptitude and neural factors may have gone undetected due to the modest power of this study.

Studying individual differences can provide insight into both variations across individuals and the neurocognitive process itself. Here, we observe that individuals with a higher executive aptitude are associated with a strategy that relies more on anterior PFC regions during semantic access task, which in turn was associated with greater recruitment of traditional semantic regions. This

provides meaningful insight into 'semantic' processes by emphasising the link between traditionally executive and semantic processes in access to complex semantic information. This link between executive and semantic processing may highlight differences in learning strategies that influence knowledge acquisitions and potentially point towards educative interventions that may improve proficiency. At the same time, identifying substrate that led to better or worse performance may provide a basis for understanding changes that can occur over the lifespan or in the neurodegenerative cases such as Alzheimer's disease, both in its full and prodromal form.

In this work, we examined the neural differences that distinguish individuals with higher or lower capacity for semantic knowledge. Results indicate that during active access to complex factual knowledge, individuals with higher executive capacity relied more strongly on lateral anterior PFC, regions associated with memory-related executive functions, as well as vmPFC and the precuneus, areas implicated in semantic processing. Resting-state analysis verified the importance of anterior PFC in semantic processing, with measures of semantic aptitude predicting functional connectivity between the left anterior PFC and the precuneus at rest. These results highlight the importance of coordination between memory-related executive processes in the prefrontal cortex and default-mode regions involved in internalised cognition in the efficient processing of factual information. By contrast, lower semantic aptitude was associated with increased activation of the posterior dmPFC, and decreased connectivity between this region and left anterior PFC predicted better semantic aptitude. This suggests that reliance on semantic control operations within posterior dmPFC may lead to a lower capacity for processing more complex factual information.

## Data availability
The datasets generated during and/or analysed during the current study are available from the corresponding author on reasonable request.

## Code availability
The codes and scripts generated during and/or analysed during the current study are available from the corresponding author on reasonable request.

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

## Acknowledgements
The project was funded by the European Research Council (ERC) grant CRASK - Cortical Representation of Abstract Semantic Knowledge, under the European Union's Horizon 2020 research and innovation program (grant agreement no. 640594). MRI scanning was supported by funding from the Caritro Foundation, Italy.

## Author contributions
Original study concept: S.L.F. Research design: S.L.F., S.U. and G.R. Stimuli and Experiment implementation: G.R. and S.U. Data analysis: G.R. and S.L.F. Manuscript drafting: G.R. and S.L.F. Manuscript review & editing: all authors.

## Competing interests
The authors declare no competing interests.
