## [Peer Review File · Communications Biology]

Reviewers' comments:

Reviewer #1 (Remarks to the Author):

This paper seeks to address individual differences in functional connectivity as they relate to semantic knowledge measured with a combination of standardized (e.g., WAIS) and custom measures. Task-based analyses were used to generate-specific ROIs which were then fed into the connectivity analyses. They find a set of regions spanning default mode (e.g., precuneus, retrosplenial cortex), frontal control (e.g., vm and dmPFC), memory (hippocampus, parahippocampal cortex), and language areas (e.g., fusiform gyrus, IFG, middle temporal gyrus) are implicated in access to semantic knowledge. Connectivity analyses revealed clustering between regions implicated in functional processing of similar categories of information (e.g., people, academic facts, etc.). Regions within PFC showed the strongest correspondence of performance on semantic knowledge tasks.

This manuscript is clearly written and uses methodologically appropriate analyses. Although, ROIs did not come from an independent sample, they were identified from a distinct phase of the task and then used in the connectivity analyses. The findings seem like they would be of interest to a relatively specialist audience interested in understanding resting state profile affiliations that may inform semantic processing.

A major issue that dampens my enthusiasm for the paper is that they fail to include control analyses that show the specificity of the regional associations with semantic processing per se. I do not intend to suggest that a region such as PFC only is implicated in semantics, but, rather, I would like them to control for other possible explanations. For instance, can they show that effects are not being driven more generally by memory search or even cognitive effort?

In part, this concern is related to issues with these standardized cognitive tasks. While they remain helpful diagnostic instruments, we no longer believe them to be process pure measurements.

Furthermore, they perpetuate socioeconomic, racial, and ethnic biases. At the very least, these issues should be thoroughly considered in the Discussion.

Methodologically, I have a few areas that would benefit from clarification – firstly, while N=43 would be a relatively large sample for group-level effects, it is much smaller than would be ideal to show individual differences. Was an a priori power analysis conducted that might help allay these concerns? Furthermore, was there adequate variability in the tasks measured? They noted a pre-testing phase for the WAIS, but selection for inclusion in the MRI sample was unclear.

A few other methods details that I think they'd benefit from including:

- Number of words in the experimental sentences (these were reported for the control sentences, and it would be nice to be able to compare across)
- Average number of trials for each of the nine possible bins
- Whether any of the "compcor" or frame displacement measures from CONN were used
- Were ROIs defined at the group or individual level?
- How was the "six most significant category-selective peaks" criteria selected?
- Handedness of participants (especially given lateralization of effects)

I am confused about the logic of how the composite measures were computed. It seems to me that summing across subscales and including a z-scored summary value would be sufficient. Why did they authors multiple by the number of subscales when calculating the overall composite? Relatedly, I do not follow the "omnibus semantic" and "omnibus executive" regression analyses – can they explain this more clearly?

A final major issue is the framing of "semantic aptitude". As the authors state in their introductory, our knowledge of the world is continuously growing and changing. Yet, I worry that by framing the paper around correlates of individual differences in "aptitude" that this frames this as a fixed skill of finite capacity. I think there is an additional onus on all of us in the MRI community to think carefully about how our work can be misconstrued, especially when trying to make claims about individual differences

in what is likely a homogeneous, non-representative sample using inherently biased neuropsychological measures. I suspect this issue could be handled by reframing of the paper and selecting a term other than "aptitude".

Minor issues include:

1. The abbreviation pMTG is repeated between lines 64 and 65
2. Labels in several of the figures are too small to be legible – e.g., Figs 1 and 3B
3. Why do they specifically focus on reward circuitry without talking about why this might be implicated in semantic processing? If they think this is interesting and can meaningfully help to understand the results then there should be greater treatment of this in the Discussion. It seems like this group has found these regions to be involved previously – do they have speculations as to why?
4. The statistical threshold in Fig 2 appears to be too lenient because it looks like the activation extends into white matter (corpus callosum – see bottom right brain section).
5. Why were "unknown" trials included (see line 358)
6. Regions enumerated in the text, starting on line 360, may be better solely listed in the accompanying table. Currently, this text simply reads as a list
7. What is the justification for the more lenient threshold for the individual differences during "semantic access" analyses?

Reviewer #2 (Remarks to the Author):

Rabini, Ubaldi, and Fairhall have studied how individual differences in task-based and resting-state connectivity predict semantic abilities. They report that distinct regions are recruited when accessing information pertaining to people, places, objects, and scholastics. While the findings are interesting, my enthusiasm for this manuscript is dampened by the lack of clarity in communicating the research and the significance of the work. Please find below my comments/questions:

1. A clearly stated definition of semantic knowledge/capacity in the introduction before diving into the existing literature would be helpful. While components of such a definition are present in the first paragraph, it might not be clear to the readers if they're not explicitly searching for it.
2. How were the questions used in the general knowledge task determined? In a typical population, what percentage of the questions should we expect individuals to know the answers to? How were the structures of the questions determined (i.e. "If I am in Cappadocia, what country am I in?" Vs. "What country is Cappadocia in?")? Were these questions validated in terms of understanding of what the question was asking, suitability to different populations (i.e., males vs females, younger vs older adults, etc) prior to their use in the study in any way? Please include additional information (either in the Methods section or the Supplementary material that includes the list of questions) that provides more information.
3. In Figure 3B, the regional labels are hard to read and it's difficult to clearly see which connections are being depicted in the chord plot. A different color-scale that's more discriminable across the range might be preferred.
4. Clusters depicted in Figure 4 seem largely disjointed - what were the voxel thresholds used to identify clusters that positively and negatively related to semantic knowledge? Was any smoothing applied to the clusters after identification to create more spatially constrained clusters?
5. Within the methods section, several acronyms are used without prior definition/explanation of what they represents (i.e., AC/PC, ART, RFX).
6. On a more abstract note, there's no clear explanation or discussion or the overall goal/value of this work. There's a difference between studying a given relationship just because you can versus studying it because it is likely to have some sort of impact whether clinically or in our understanding of the

brain. While there is clear value in this work, authors do not articulate the overall significance of the work in their manuscript.

Response to Reviewers

We would like to thank the reviewers both for their enthusiasm towards this work and their helpful comments and suggestions. We address each point below and feel that in so doing we have significantly improved the quality of the manuscript.

Reviewers' comments:

Reviewer #1 (Remarks to the Author):

This paper seeks to address individual differences in functional connectivity as they relate to semantic knowledge measured with a combination of standardized (e.g., WAIS) and custom measures. Task-based analyses were used to generate-specific ROIs which were then fed into the connectivity analyses. They find a set of regions spanning default mode (e.g., precuneus, retrosplenial cortex), frontal control (e.g., vm and dmPFC), memory (hippocampus, parahippocampal cortex), and language areas (e.g., fusiform gyrus, IFG, middle temporal gyrus) are implicated in access to semantic knowledge. Connectivity analyses revealed clustering between regions implicated in functional processing of similar categories of information (e.g., people, academic facts, etc.). Regions within PFC showed the strongest correspondence of performance on semantic knowledge tasks.

This manuscript is clearly written and uses methodologically appropriate analyses. Although, ROIs did not come from an independent sample, they were identified from a distinct phase of the task and then used in the connectivity analyses. The findings seem like they would be of interest to a relatively specialist audience interested in understanding resting state profile affiliations that may inform semantic processing.

We thank the review for these positive comments.

1. A major issue that dampens my enthusiasm for the paper is that they fail to include control analyses that show the specificity of the regional associations with semantic processing per se. I do not intend to suggest that a region such as PFC only is implicated in semantics, but, rather, I would like them to control for other possible explanations. For instance, can they show that effects are not being driven more generally by memory search or even cognitive effort?

Thank you for pointing out this issue. We attempted to mitigate this concern by employing the comparison of successful versus unsuccessful access as the basis for our analyses. There are features of this contrast that make it less likely that effects are being driven by general cognitive effort or memory search. Reaction times suggest that general cognitive effort does not contribute, as RTs are actually longer in the unsuccessful access condition. Memory search is presumably present in both trial types, to some extent. However, while the longer reaction times suggest that memory search might persist for longer in unsuccessful access trial it remains unknown whether this is balanced or more extensive in one or the other conditions.

To address this issue better, we have now added the following text to the discussion:

"This contrast was chosen to better control for broad effects of task and focus on aspects of cognition most closely associated with semantic access. Analysis of reaction times suggests that

general effects of cognitive load were absent from this contrast, as RTs were longer in unsuccessful access trials. While these RT differences make it less likely that other non-specific factors, such as general memory search, are contributing to this contrast, such contributions cannot be excluded.”

- pages 21-22

2. In part, this concern is related to issues with these standardized cognitive tasks. While they remain helpful diagnostic instruments, we no longer believe them to be process pure measurements. Furthermore, they perpetuate socioeconomic, racial, and ethnic biases. At the very least, these issues should be thoroughly considered in the Discussion.

We fully agree that there is a potential for the overinterpretation, and even misuse of, standardised measures and their application to individual differences. We have now included a section in the discussion where we note the limitations of this test and caution against overinterpretation.

“These divergent findings highlight potential considerations relating to the measures used to quantify ‘semantic aptitude’. In the present study we use a combination of standard psychometric measures and task performance as a proxy for semantic aptitude, but it should be noted that these indices do not reflect pure measurements of the underlying cognitive process and are known to be biased as a result of cultural background and socioeconomic status (Reynolds and Ramsay, 2003; Walker et al., 2009). Interpretation of the present results should be made within the limitations of the psychometric scales used to quantify semantic aptitude.”

[see also response to major comment #7 below]

- Page 25

3. Methodologically, I have a few areas that would benefit from clarification – firstly, while N=43 would be a relatively large sample for group-level effects, it is much smaller than would be ideal to show individual differences. Was an a priori power analysis conducted that might help allay these concerns? Furthermore, was there adequate variability in the tasks measured? They noted a pre-testing phase for the WAIS, but selection for inclusion in the MRI sample was unclear.

Apologies that this was unclear. The motivation for the pre-testing phases was to ensure adequate variability in the measures. The 73 participants were selected to allow us to sample evenly across the distribution of general information scores, a strategy that was chosen to attain adequate sensitivity of the measure. We have now clarified this in the manuscript

‘This approach was chosen to attain an adequate spread in the distribution of available scores and increase the sensitivity of brain/behaviour measures.’

- page 4

We did not perform an a priori power analysis but we have now noted that the lack of power may have led to a failure to detect other effects, noting that:

“[...] additional relationships between aptitude and neural factors may have gone undetected due to the modest power of this study.”

- Page 25

**4. A few other methods details that I think they’d benefit from including:
• Number of words in the experimental sentences (these were reported for the control sentences, and it would be nice to be able to compare across)**

Thank you for pointing out this omission. The experimental sentences were also 8-12 words (the control stimuli were constructed to match this). This has now been indicated in the text (page 5).

- **Average number of trials for each of the nine possible bins**

There were 12 control trials per run (48 in total). This has now been reported in the methods section (page 5).

The number of trials in the experimental conditions is contingent on the proportion of known and unknown responses. This was about 50% for each of the domains meaning that the average number of trials in each of the experimental bins ranged from 25-35. To make this information available to the reader, we now report both the number of and percent of 'known' trials in the relevant section of the results.

“Specifically, post-hoc tests indicated that the mean number of known facts for People (25.6/60, 42.7%) and Places (26.0/60, 43.4%; no significant differences) were lower than known facts for Object (30.1/60, 50.1%; vs. people, $t_{(40)} = 3.18$, $p = .011$, vs. places, $t_{(40)} = 2.87$, $p = 0.029$) and Scholastic Domains (30.3/60, 50.5%; vs. people, $t_{(40)} = 3.39$, $p = .006$; vs. places, $t_{(40)} = 3.08$, $p = .015$).”

- Page 10

- **Whether any of the “compcor” or frame displacement measures from CONN were used**

Neither component based or frame-displacement based denoising steps were implemented (only ART, CSF/WM regressors and temporal filtering were used).

- **Were ROIs defined at the group or individual level?**

ROIs were defined at the group level, this has now been specified in the text (page 10).

- **How was the “six most significant category-selective peaks” criteria selected?**

This was based upon the t-value and refers to the local maximas reported in table 2. This has now been clarified in the text (page 9).

- **Handedness of participants (especially given lateralization of effects)**

All participants were right-handed native Italian speakers (page 4)

5. I am confused about the logic of how the composite measures were computed. It seems to me that summing across subscales and including a z-scored summary value would be sufficient. Why did they authors multiple by the number of subscales when calculating the overall composite?

Thank you. We appreciate that this process seems a little convoluted. The initial averaging of the Information, Vocabulary, and Similarities was implemented via the WIAS-IV process, which is non-linear in nature. As mentioned in the text, the output of this, now single, measure was multiplied by three so that its weight in the weighted average reflect the three independent measures that compose it.

We have now clarified this in the text”

“Information, Vocabulary, and Similarities subscales were combined and weighted according to WAIS-IV criteria (which is non-linear; Wechsler, 2008). A weighted average was then performed including this measure, the percentage of answers known from the fMRI test, the Semantic Encoding test and the Verbal Fluency tests (specifically, the composite Information/Vocabulary/Similarity measure was weighted by 3 to reflect the number of initial tests, while the other scales were weighted by 1).”

- Page 8

6. Relatedly, I do not follow the “omnibus semantic” and “omnibus executive” regression analyses – can they explain this more clearly?

Thank you for drawing our attention to the lack of clarity in our explanation of this analysis.

Rather than using multiple scales we collapsed like scales that cluster together across individuals (see dendrogram in figure 1) to reduce the number of initial regressions (we later decompose these measures in post hoc tests to investigate the role of subscales).

Then, at each voxel of the brain, we used difference in individuals' scores on the omnibus semantic (or executive) scale to predict the magnitude of the difference between the BOLD-derived beta.

We have now added the following text to the manuscript:

"To reduce multiple comparisons and focus initially on broader aspects of cognition, we collapsed like scales that cluster together across individuals (see dendrogram in figure 1) into two composite measures (see methods)."

- Page 8

and

"Specifically, at each voxel of the brain, we used difference in individuals' scores on the omnibus semantic or executive scale to predict the magnitude of the difference between the BOLD-derived beta response for known versus unknown trials."

- Page 17

7. A final major issue is the framing of "semantic aptitude". As the authors state in their introductory, our knowledge of the world is continuously growing and changing. Yet, I worry that by framing the paper around correlates of individual differences in "aptitude" that this frames this as a fixed skill of finite capacity. I think there is an additional onus on all of us in the MRI community to think carefully about how our work can be misconstrued, especially when trying to make claims about individual differences in what is likely a homogeneous, non-representative sample using inherently biased neuropsychological measures. I suspect this issue could be handled by reframing of the paper and selecting a term other than "aptitude"

Thank you again for raising these considerations. We have now specifically drawn attention to the non-representative nature of our sample and cautioned about drawing conclusions about the general population.

"At the same time, it should be noted that our sample reflects a culturally and ethnically homogenous population and the external validity of the present results to differing populations remains untested [...]"

- Page 25

We had found it hard find an unloaded term to refer to differences in semantic ability. We landed on semantic aptitude as being relatively neutral, with alternative phrases being more/equivalently loaded, overly cumbersome or inaccurate. Rather than replace this term with another that may have different but equivalent shortcomings, we have highlighted that this is a convenience term and operationally defined it early in the manuscript, adding the following text:

"Here, we have used semantic aptitude as a convenience label that we operationally define as the observed performance on standardized and custom tests of general factual knowledge and does not imply that this is an unchangeable aspect of cognition that is not altered and shaped by culture or even personal proclivity".

- Page 2

Minor issues include:

1. The abbreviation pMTG is repeated between lines 64 and 65

Thank you. The additional pMTG has been removed.

2. Labels in several of the figures are too small to be legible – e.g., Figs 1 and 3B

These have both been corrected.

3. Why do they specifically focus on reward circuitry without talking about why this might be implicated in semantic processing? If they think this is interesting and can meaningfully help to understand the results then there should be greater treatment of this in the Discussion. It seems like this group has found these regions to be involved previously – do they have speculations as to why?

One can imagine that that knowing the answer to a question is associated with a feeling of accomplishments (trivia-based games are very popular). Although, this is a highly speculative.

We now have included a mention of this possibility while emphasising that this activation is not predictive of semantic capacity (the scope of this work).

“The reason for the involvement of reward circuitry in successful semantic retrieval is uncertain but may reflect the feeling of accomplishment associated with knowing the answer to a challenging question. Importantly, we did not see evidence that the extent of recruitment in these regions correlated with overall semantic aptitude.”

- Page 22

4. The statistical threshold in Fig 2 appears to be too lenient because it looks like the activation extends into white matter (corpus callosum – see bottom right brain section).

The threshold is $p < .001$ (in line with the conventions widely used in the literature). We agree that there seems to be some blurring due to smoothing or imperfections in across subject alignment.

5. Why were “unknown” trials included (see line 358)

This was simply for consistency with our previous study. Importantly, the results were not meaningfully different when only known trials were considered. We have added the following text to the manuscript.

“Known and unknown trials were used for consistency with previous work (Ubaldi et al, 2022) and results did not qualitatively differ if only known trials were used.”

- page 14

6. Regions enumerated in the text, starting on line 360, may be better solely listed in the accompanying table. Currently, this text simply reads as a list

Thank you for pointing this out. We have now removed all but the first sentence of that paragraph.

7. What is the justification for the more lenient threshold for the individual differences during “semantic access” analyses?

It should be noted that the more lenient threshold applied to the ‘descriptive’ voxel-level threshold, inference occurs at the cluster level and inferential thresholds remains as strict as other analyses (i.e. $p < .05$, corrected).

Having said that, there is evidence suggesting that this form of threshold can increase the chance of type 1 errors with random-field-theory based familywise error correction (Eklund et al, 2016;

Flandin & Friston, 2019). However, we believe that some leniency is acceptable in this case as this approach was used to localise more extensive regions for subsequent analysis and verification with resting state data. We have attempted to clarify these considerations in the new text:

“This lower threshold allows us to detect voxels whose involvement is more subtle while keeping statistics at the inferential level consistent at $p < .05$ corrected. However, this lower initial threshold may be associated with an increased probability of false positives (Eklund et al, 2016; Flandin & Friston, 2019) and it is therefore important that results within the identified regions are validated and verified in the subsequent resting state analysis (see next section).”

- Page 17

Reviewer #2 (Remarks to the Author):

Rabini, Ubaldi, and Fairhall have studied how individual differences in task-based and resting-state connectivity predict semantic abilities. They report that distinct regions are recruited when accessing information pertaining to people, places, objects, and scholastics. While the findings are interesting, my enthusiasm for this manuscript is dampened by the lack of clarity in communicating the research and the significance of the work. Please find below my comments/questions:

1. A clearly stated definition of semantic knowledge/capacity in the introduction before diving into the existing literature would be helpful. While components of such a definition are present in the first paragraph, it might not be clear to the readers if they're not explicitly searching for it.

Thank you for this advice. We have now included the following passage in the introduction.

“Semantic memory includes a broad range of knowledge that is unrelated to the experience of a specific individual. This extends from the meaning of words, to knowledge about basic concepts ('dog': a four legged domesticated mammal), to complex encyclopaedic knowledge about the world (Tulving, 1972). While knowledge about basic concepts is relatively uniform across individuals, quantifiable variations exist in our semantic capacity, in terms of our ability to acquire, retain and reliably access complex forms of encyclopaedic factual knowledge, as measured by standardised tests such as the general information scale of the Weschler Adult Intelligence Scale (WAIS).”

- Pages 1-2

2. How were the questions used in the general knowledge task determined? In a typical population, what percentage of the questions should we expect individuals to know the answers to? How were the structures of the questions determined (i.e. “If I am in Cappadocia, what country am I in?” Vs. “What country is Cappadocia in?”)? Were these questions validated in terms of understanding of what the question was asking, suitability to different populations (i.e., males vs females, younger vs older adults, etc) prior to their use in the study in any way? Please include additional information (either in the Methods section or the Supplementary material that includes the list of questions) that provides more information.

Questions were derived from our previous study (Ubalid et al, 2022, JNeuro). These questions were selected to be challenging (this prior sample reported 46% of the answers to be known) and

with sufficient variation in item accuracy across participants (mean correlation between subjects in items-known, $r=0.21$).

The wording of the questions was designed to have relative uniformity of sentence lengths across stimuli and domains.

The questions have not been validated across a wide range of samples (gender, age) although results seem to be consistent across samples drawn from within our northern Italian population. On average, 47% of the items were reported as known by current sample (compared to 46% in the earlier study). The scale demonstrated good convergent validity with Verbal subtests of the WAIS (Information, Vocabulary, Semantic Fluency, Similarities; c.f. figure 1), that was maximal with the General Information scale ($r=0.46$), as well as discriminate validity with respect to the Performance tests (Digit Span, Symbol Search, Symbol Coding), collectively indicating a reasonable construct validity for this population.

We have now included:

A caution against extrapolating to other samples

“At the same time, it should be noted that our sample reflects a culturally and ethnically homogenous population and the external validity of the present results to differing populations remains untested [...]”

- Page 25

Description of the construct validity in text near figure 1

“The questions used in the fMRI study are not designed to be a standardised test and have not been validated across a wide range of samples (gender, age). However, within our northern Italian population, the scale demonstrated reasonable convergent validity with Verbal subtests of the WAIS (Information, Vocabulary, Semantic Fluency, Similarities; c.f. figure 1), that was maximal with the Information scale ($r=0.46$). At the same time, performance on the fMRI task demonstrated discriminate validity with respect to the Performance tests (Digit Span, Symbol Search, Symbol Coding). Collectively, this suggests that the fMRI task has an acceptable construct validity for this population.”

- Page 12

3. In Figure 3B, the regional labels are hard to read and it's difficult to clearly see which connections are being depicted in the chord plot. A different color-scale that's more discriminable across the range might be preferred.

We have altered the figure to increase legibility (below). Different colour-scales were considered but without using blue tones, that are classically associated with negative/anti-correlated connectivity between regions (which may produce confusion), this one was the most discriminable.

A. Task-Based Knowledge-Domain-Selectivity

B. Resting State Functional Connectivity

4. Clusters depicted in Figure 4 seem largely disjointed - what were the voxel thresholds used to identify clusters that positively and negatively related to semantic knowledge? Was any smoothing applied to the clusters after identification to create more spatially constrained clusters?

The extent threshold that survives correction for multiple comparisons is determined by the data (via random field theory; Friston et al, 1994, Human Brain Mapping). In practice for these analyses the threshold was equivalent to <500 voxels. No spatial smoothing was applied after identification. We have now included the extent threshold in the figure caption.

5. Within the methods section, several acronyms are used without prior definition/explanation of what they represents (i.e., AC/PC, ART, RFX).

Thank you. These acronyms have now been clarified.

6. On a more abstract note, there's no clear explanation or discussion of the overall goal/value of this work. There's a difference between studying a given relationship just because you can versus studying it because it is likely to have some sort of impact whether clinically or in our understanding of the brain. While there is clear value in this work, authors do not articulate the overall significance of the work in their manuscript.

Thank you for this prompt. We have favoured conservatism rather than getting to speculative about future potential implications of this basic research. Broadly, we were driven towards this work because we felt that understanding which neurocognitive strategies were associated with better performance could shed better light on the neurocognitive mechanism that underlies this complex form of knowledge as well as the cognitive strategies that may lead to better educational performance, while at the same time laying the individual differences groundwork for understanding how changes in this complex form of semantic knowledge change as we age or in neurodegenerative disease such as Alzheimer's Disease.

We have now added the following to the discussion:

“Studying individual differences can provide insight into both variations across individuals and the neurocognitive process itself. Here, we observe that individuals with a higher executive aptitude are associated with a strategy that relies more on anterior PFC regions during semantic access task, which in turn was associated with greater recruitment of traditional semantic regions. This provides meaningful insight into ‘semantic’ processes by emphasising the link between traditionally executive and semantic processes in access to complex semantic information. This link between executive and semantic processing may highlight differences in learning strategies that influence knowledge acquisitions and potentially point towards educative interventions that may improve proficiency. At the same time, identifying substrate that led to better or worse performance may provide a basis for understanding changes that can occur over the lifespan or in the neurodegenerative cases such as Alzheimer’s disease, both in its full and prodromal form.”

Giuseppe Rabini, PhD

University of Trento

Center for Mind/Brain Sciences (CIMEC)

Corso Bettini, 31

38068 - Rovereto, Italy

REVIEWERS' COMMENTS:

Reviewer #1 (Remarks to the Author):

The authors seem to have thoroughly addressed the concerns of the reviewers. They took particular care with the issues raised around biases in measures which I appreciated.

I still remain confused about their claim that successful vs. unsuccessful access fully controls for other differences (I actually think the fact that there's RT differences hurts their argument), however they do now acknowledge issues related to interpretation in the Discussion and I think that is likely sufficient.

There do seem to be some changes to the RT values on page 11 that I could not link to any reviewer comments. Certainly if they noticed an error, this should be fixed however I didn't see any notes in their response letter to indicate that I should have expected this change. I just want to verify that we are now viewing the correct values here.

Reviewer #2 (Remarks to the Author):

Authors have addressed all of my concerns.

Response to Reviewers

We would like to thank the reviewers both for their enthusiasm towards this work and their helpful comments and suggestions. We address each point below and feel that in so doing we have significantly improved the quality of the manuscript.

Reviewers' comments:

Reviewer #1 (Remarks to the Author):

The authors seem to have thoroughly addressed the concerns of the reviewers. They took particular care with the issues raised around biases in measures which I appreciated.

I still remain confused about their claim that successful vs. unsuccessful access fully controls for other differences (I actually think the fact that there's RT differences hurts their argument), however they do now acknowledge issues related to interpretation in the Discussion and I think that is likely sufficient.

There do seem to be some changes to the RT values on page 11 that I could not link to any reviewer comments. Certainly if they noticed an error, this should be fixed however I didn't see any notes in their response letter to indicate that I should have expected this change. I just want to verify that we are now viewing the correct values here.

Apologies. We indicated the changed RTs in the track-changes but did not specify why these changed.

Three RT's values had been mis-transcribed in the original submission. Importantly, all statistics and the descriptions of the direction of effects were correct, only the written magnitude of the RT in the initial submission reported erroneously and has now been corrected.